# The Need to Maintain Sustainability in the Dynamic Anthropogenic Changes in the Natural Landscape of the Bay of Pomerania in Poland

**Katarzyna Krasowska**  and **Zbigniew W. Paszkowski** *

Department of Architecture, West Pomeranian University of Technology, ul. Zolnierska 50,
71-210 Szczecin, Poland
* Correspondence: prof.paszkowski@gmail.com; Tel.: +48-601704454

**Abstract:** This article presents a research study into the dynamics of negative changes to the almost untouched coastal landscape of the Bay of Pomerania on the south-western coast of the Baltic Sea, taking into consideration the impact of the spatial development of the port of Świnoujście and of the development of leisure facilities in the small fishing villages of the Baltic coast. The authors highlight the natural landscape dynamics resulting from the topography, the aggression of the Baltic Sea against the beaches and sandy moraine hills, the dynamics of phyto-biological development in the endemic natural environment, and the dynamics of anthropogenic landscape changes resulting from the oversized spatial development of both residential and industrial buildings, as well as intermodal and port infrastructure development. The study analyzes natural values and their importance for both health and recreational purposes and shows the disproportional impact of economic/industrial development on the study area in recent times. The research was undertaken to verify the necessity of maintaining sustainability in the anthropogenic and industrial development undertaken in those areas, in order to protect sensitive ecological areas and to provide environmental compensation for the negative landscape changes already produced.

**Keywords:** landscape; dynamics of landscape changes; sustainable development; port development; Świnoujście



## 1. Introduction

The landscape that surrounds us is characterized by a high degree of complexity. Its natural and anthropogenic elements interact with each other, changing over time and space. The landscape, both natural and cultural, shaped over the centuries, is currently subjected to processes of rapid change, resulting from the development of civilization and social–political changes. A better understanding of the regularities governing changes in land use, especially in areas characterized by expansive dynamics of economic development, is now a key challenge for the landscape [1]. Changes in land cover strongly affect the stability of ecosystems, their biodiversity and the ability to provide services, resulting in a direct impact on the level of well-being of the human population [2]. It is not only the course of these changes itself that is worth knowing; in order to understand the mechanisms responsible for landscape change, it is necessary to identify the driving forces leading to landscape transformation and the spatial determinants shaping the mosaic of landscape/land cover (LC) pattern [3]. In today's world, which more and more often irreversibly interferes with the shape of the landscape, it is imperative to create the concept of landscape management optimization and to establish the limits of permissible changes.

Port cities, by necessity, have a close relationship with the natural environment. Shifting water lines, rising sea levels, and the threat of flooding are among the many natural challenges that shipping companies and port city governments have had to tackle for centuries. Given their dependence on the natural environment to allow for shipping, which

is their main economic driver, port cities provide a good case study for the examination of long-term economic development, urban wealth creation and resource management for the benefit of urban leaders, and the creation of urban sustainability and resilience, issues that have come to the fore in recent decades. Sustainability, as defined in multiple publications since the Brundtland Commission report in 1987, implies a balance of economic development, natural environment and social equity, with a focus on the well-being of current and future generations [4]. Since the turn of the millennium, the quest of European port authorities for strategies that secure their competitiveness and "license to operate" has become increasingly complex. Sustainability agendas are challenging port authorities around the world to find ways to use port assets more efficiently and productively in economic, social, as well as environmental terms [5].

The research area is covered by a transportation system that features all branches and types of transport. One of the biggest Polish ports in the Baltic Sea is located here—the integrated Port of Szczecin-Świnoujście. The location of the port in Świnoujście, which has a lot of benefits due to the proximity of international transportation routes, is key to the observed as well as planned development of intermodal infrastructure, which causes changes in the features of existing areas and expands them at the cost of natural land (forests, agricultural areas and grassland). The dynamic development of investments in the transportation sector that is both observed and planned in the analyzed area has a negative impact on the natural environment in several ways—the annexation of forested and agricultural areas, environmental pollution, the interruption of ecological corridors, the destruction of endemic species of flora and fauna, and the interruption of the ecological balance [6,7].

In order to better understand the causes of current land use, its diversity and historical circumstances, as well as to create future development scenarios, it is necessary to conduct research in regions with particular sensitivity to landscape change. To achieve research results that have more than simply a local impact and to demonstrate irregularities, which can be assumed to be features of general irregularities, the research area should be sufficiently extensive (in landscape scale) and representative of its specific type of landscape. A credible analysis of the relationship between variables in the landscape system also requires the examined landscape qualities to be as varied as possible in terms of the time and space of the research [8]. The aim of this article is:

- to discover, monitor and assess the dynamics of qualitative and spatial changes in the valuable coastal landscape of the Bay of Pomerania on the south-west coast of the Baltic Sea.
- to encourage sustainable, synergistic landscape management as one of the most important factors in complex, multi-sectoral, multi-spatial planning at local, regional and national levels.

## 2. Materials and Methods

### 2.1. Area-Time Delimitation of the Research

The research was based in the area of the south-west coast of the Bay of Pomerania on the Baltic Sea, located in the north-west part of the Voivodeship of West Pomerania. It includes: parts of the islands of Uznam and Wolin; the area of the port city of Świnoujście located at the mouth of the Świna, with its extensive holiday resort district at the Baltic seaside; the western part of the Odra River, together with the area of the Świnoujście commune, the area of the recreational and spa town of Międzyzdroje, the Baltic coastal strip, the western part of Wolin National Park and the northern part of the Szczecin Lagoon, constituting the estuary of the mouth of the Odra River (Figure 1). This article presents an analysis of the changes that have taken place in the landscape since 2004, i.e., the date of Poland's accession to the European Union. This date was the turning point for increased international tourist traffic, mainly from Germany and Scandinavia, to holiday resorts located on the southern coast of the Baltic Sea. It was also the period of implementation of infrastructure projects supported by EU funds, aimed at levelling up to European standards

in the field of intermodal transport, safety and quality of life, as well as environmental protection rules [9–11].

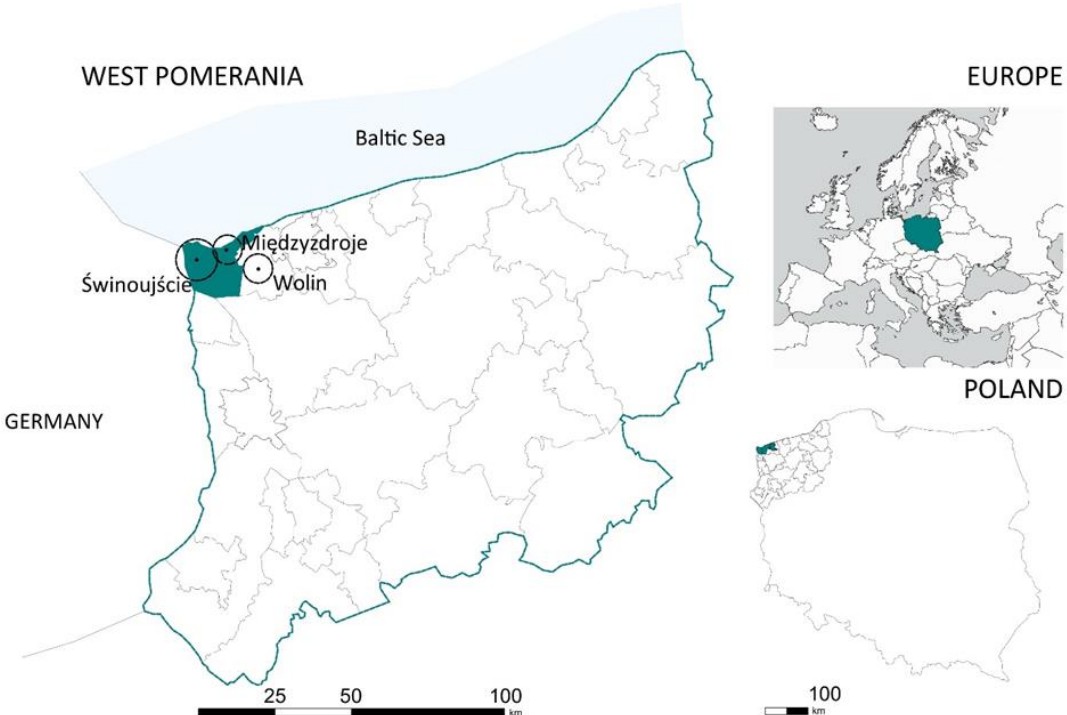

**Figure 1.** Location of the study area—The Bay of Pomerania (Poland). The author's original work made using the ArchiCad program.

*2.2. Methods Applied*

This study is based on the analysis of statistical data: quantitative analysis of allocated accommodation in the area, analysis of the area of forest land, analysis of the area of legally protected areas, analysis of the number of facilities providing accommodation to tourists, analysis of the length of bicycle paths, analysis of the urbanization level of the area: number of new apartments, analysis of the area of new covered and non-residential facilities (Table 1); the analysis of documents: Landscape Resolution of the City of Świnoujście [10], documentation on the Assumptions for the construction of a container terminal in Świnoujście, development strategy for the Zachodniopomorskie Voivodeship for the years [11], Regulation of the Minister of Infrastructure on the border of the sea port in Świnoujście [12], the presentation of projects, and our own analyses and observations carried out in the area of the research, allowing for full recognition of the issues of the area in question, in particular with regard to changes taking place in the landscape [13]. Statistical data show a large increase in tourist traffic in the West Pomeranian Voivodeship; in 2009–2019, the number of overnight stays in the research area almost doubled (Figure 2). Since 2005, a total of over 450 thousand square meters of usable area has been designated for facilities intended for tourist functions (according to BDL [14].

The landscape is an open system, and its changes result from the influence of natural environmental factors (growth and biological degradation), climatic events (storms, winds, tornadoes, floods, etc.) and the influence of factors related to direct human activity or the indirect effects of this activity. Anthropogenic changes in the landscape may arise from planned activities, resulting from the implementation of economic and spatial goals, as well as from destructive activity, resulting from a lack of knowledge or empathy or as a result of deliberate actions that destroy the environment.

**Table 1.** Direction of changes in selected localities. Source: author's study based on BDL (Local Data Bank) available on 1 September 2022.

| | | Międzyzdroje | | Świnoujście | | Wolin | |
|---|---|---|---|---|---|---|---|
| | | Urban-Rural Commune | | City with Poviat Rights | | Urban–Rural Commune | |
| | | **2010** | **2019** | **2010** | **2019** | **2010** | **2019** |
| | area [hectares] | 11,438 | 11,607 | 19,723 | 20,207 | 32,746 | 32,762 |
| | Population | 6731 | 6452 | 41,475 | 40,888 | 12,547 | 12,115 |
| **environment** | legally protected areas [hectares] | 7176.30 | 5380.67 | 3051.70 | 3052.08 | 1620.90 | 1119.60 |
| | forest area [hectares] | 53.6 | 8.31 | 146.9 | 103.92 | 118.9 | 176.76 |
| | green areas [hectares] | 13.1 | 11.32 | 21.4 | 18.44 | 109.9 | 128.06 |
| **tourism** | accommodation provided to foreign tourists | 20,913 | 76,482 | 46,448 | 262,690 | 374 | 2200 |
| | tourist accommodation facilities | 42 | 68 | 80 | 103 | 16 | 12 |
| | cycling [km] | 12.5 | 4.5 | 21.2 | 37.7 | 12 | 12 |
| **urbanization** | new apartments | 2010–2019 | | 2010–2019 | | 2010–2019 | |
| | | 1173 | | 353 | | 1831 | |
| | new residential buildings [m$^2$] | 64,301 | | 117,451 | | 47,511 | |
| | new non-residential buildings [m$^2$] | 47,128 | | 257,360 | | 13,829 | |

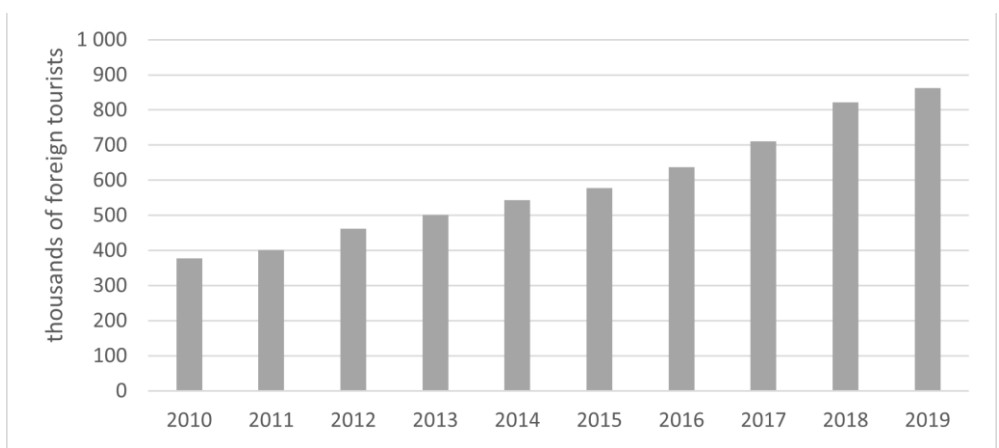

**Figure 2.** The chart shows the number of overnight stays provided to foreign tourists in the area in the years 2010–2019. Author's compilation based on GUS data (accessed on 20 March 2021).

The dynamics of landscape change refer to the process of landscape transformation, the essence of which is the relationship between the human factor and the natural environment. Monitoring the dynamics of landscape change is of particular importance in ecologically sensitive areas. Landscape can be seen as an interactive "equation" that is constantly changing through continuous processes taking place in nature [5]. The systems approach is helpful in understanding the essence of the dynamics of landscape change (Table 2).

**Table 2.** Determinants of the dynamics of landscape change. Source: author's work.

| Landscape Domain | Forces | | Indicators—Effects on the Landscape |
|---|---|---|---|
| The coast | Societal factors | - industrialization<br>- changing tourism markets<br>- urbanization | - degradation of natural resources, NATURA 2000 area of natural habitats<br>- change in the forest area<br>- limitation of the area of beaches<br>- decline in visual quality<br>- new apartment buildings in the immediate vicinity of protected zones |
| | Natural factors | - climate change<br>- storms<br>- rising sea levels | - subsidence of the waterfront<br>- the degradation of beaches and cliffs<br>- reducing the area of beaches |
| The urban core | Societal factors | - development pressures<br>- development of residential buildings | - degradation of the city space<br>- new trends in shaping buildings<br>- raising the height of buildings |
| | Natural factors | - climate change | - reducing the biologically active surface<br>- drought<br>- climate change |
| The rural land | Societal factors | - changes in agricultural techniques<br>- migration of previous farmers from villages to cities<br>- migration of city dwellers to the rural land | - degradation of natural resources, NATURA 2000 area of natural habitats<br>- increasing the density of buildings<br>- agricultural land decay for the purposes of apartment buildings<br>- urban sprawl, urbanization of rural areas |
| | Natural factors | - deforestation<br>- industrialization | - construction of wind farms<br>- reduction of agricultural land<br>- degradation of bird habitats<br>- change in the perception of rural spaces, industrial character |

The ongoing process of feedback completes the linear dependence of the input data and the results of the processes taking place [15–17]. When analyzing the dynamics of changes taking place in the landscape, one should take into account such variables as: the frequency of changes, the size of changes, the subject of changes and the time of changes (Figure 3) [18,19].

The analysis was conducted on two groups of factors that influence the dynamics of landscape changes: environmental and anthropogenic. An analysis of statistical data (big data) was carried out, in which the following indicators of the dynamics of changes were considered important—the change in the area of forests, protected areas, as well as the development of tourist traffic, development of urbanized areas and road and technical infrastructure—through a quantitative analysis of new investments, the dynamics of the anthropogenic pressure on the landscape related to the increase in tourist traffic and man-made changes: large tourist investments (apartment buildings and hotels), intermodal infrastructure, such as the construction of the S3 expressway, the expansion of the railway line, the construction of a road tunnel under the Świna River, the expansion of the port and an international ferry terminal, as well as the planned construction of an external deep-water container port in Świnoujście. During the last stage of the research, changes in the landscape resulting from the action of natural environmental factors were analyzed. The method of color marking of functional areas was used to graphically present the results of spatial correlation studies (Figure 4) [20,21].

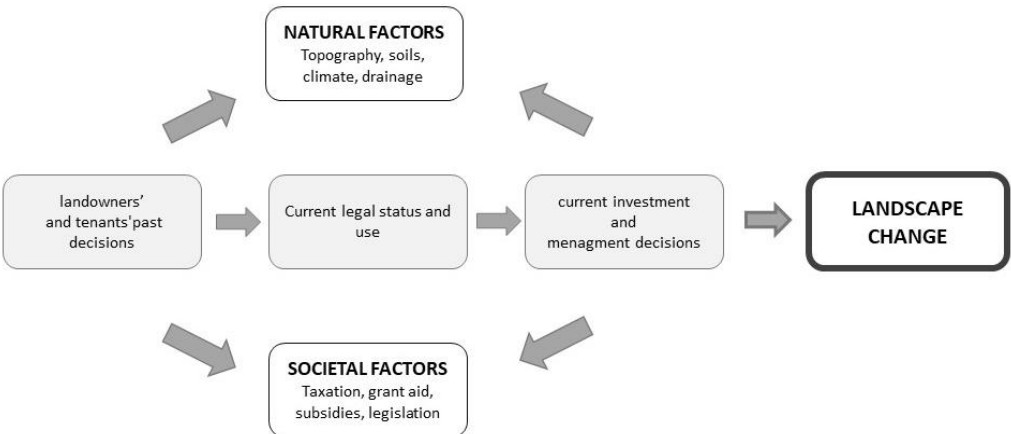

**Figure 3.** A simple model of landscape change [18,19]. Own study based on Selman (p. 27).

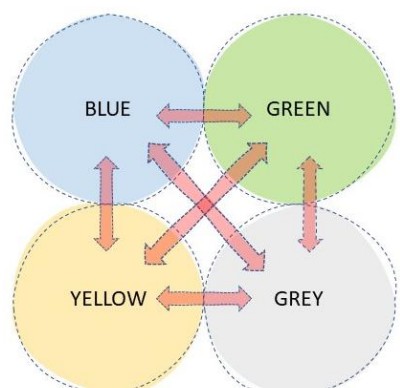

**Figure 4.** Integrated planning in the landscape of the Odra estuary at its confluence with the Bay of Pomerania on the Baltic Sea covers mutually competing areas: BLUE—coastal areas and waters, GREEN—areas protected in terms of nature, YELLOW—areas of tourist development, including buildings and complexes constituting cultural heritage, GREY—areas for the development of multimodal transport (roads, railways, ports and ferry crossings).

The principles of sustainable development, constituting the basis for the spatial and economic planning directives of the European Union, and also included in the Constitution of the Republic of Poland, were adopted as a reference point for the consideration and formulation of research conclusions. "Coastal landscapes are spaces and places of conflict, being among the most dynamic and vulnerable environments on the planet. Coastal landscapes are regulated through various policy instruments focused on competing interests and land uses at several intersecting administrative scales" [22,23].

## 3. Results

### 3.1. Societal Factors

#### 3.1.1. Tourism

Increased tourist traffic and the decreasing resources of construction areas for new facilities intended for tourist services have made investors look for new solutions and opportunities for tall and high-intensity development. The gradual change of the development paradigm, from guesthouses with several stories floors to high-rise buildings with a hotel or apartment function, has made significant changes in the seaside landscape in the research area [24].

Seaside towns on the coast of the Bay of Pomerania in Poland are increasing in their scale of development, thus, changing the paradigm of imperial Wilhelminian resorts

into Mediterranean models of multifunctional hotel complexes serving mass tourism. Increasing differences are visible when comparing the landscape of spa towns in Germany (Ahlbeck, Heringsdorf and others), located just across the Polish–German border, with the development of spa towns on the Polish coast, despite their common provenance and original uniformity of architectural character [25,26].

　　Within the research area, the dynamics of changes in the landscape shaped by man related to the increase in tourist traffic has had a huge impact on the overall condition of the landscape. In the countryside located directly on the Baltic Sea (Świnoujście and Międzyzdroje), the area of forests decreased in the period analyzed. In the area of the Międzyzdroje commune, the loss of forested areas is alarming (2004: 53.10 ha, 2019: 8.31 ha) (according to BDL, date of access: 25 April 2021) [15,27–30]. A positive result was the increase in forest areas in the Wolin Commune, despite the construction of the S3 expressway. Deforestation related to this investment was mitigated by new plantings. The reforestation planned and completed in the area of investigation was based only on native tree species: pines, oaks, birches and beeches (Figure 5).

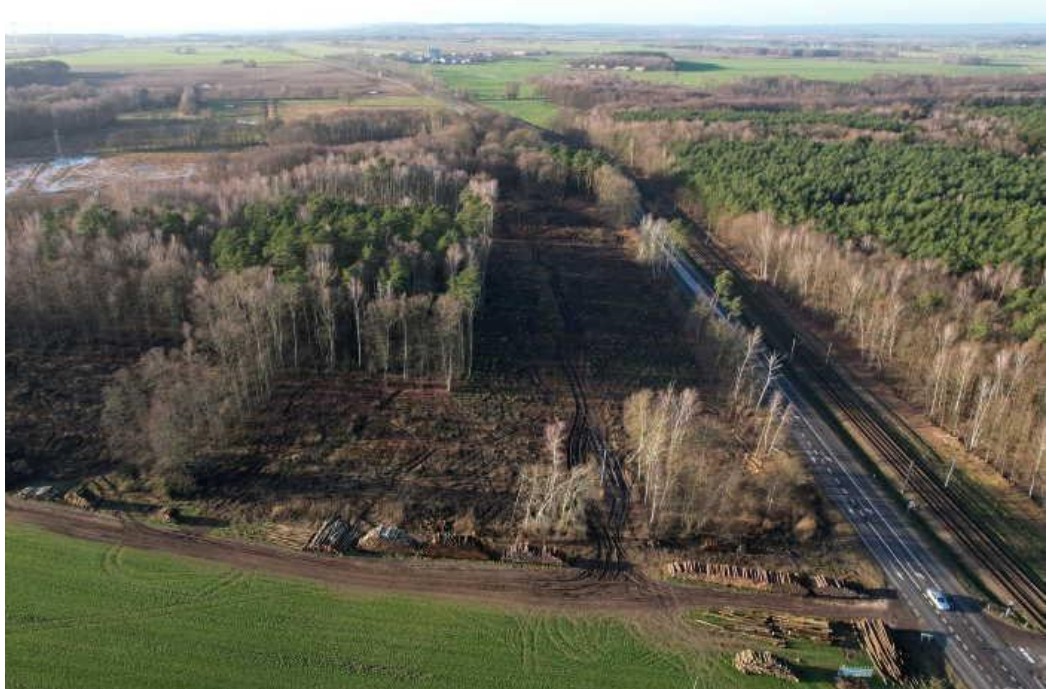

**Figure 5.** The photo shows the felling of trees alongside the effective S3 expressway in the Wolin Commune. Here, 126.32 ha of forest areas were cleared for the road; however, due to replacement plantings, the size of forest areas in this area has increased by 57.86 ha (Table 1). Source: https: //conadrogach.pl/ (accessed on 20 September 2022).

3.1.2. Transport Infrastructure

　　The construction of the S3 expressway (Figure 6), which is nearing completion, is now challenged with the task of building a difficult section of the road through the Wolin National Park, with a significant impact on an area under strict environmental protection. Hundreds of hectares of pine and oak woods are being cut down in order to provide smooth communication with the growing seaside transportation facilities and with Germany. Currently, works are also being carried out on boring a tunnel under the Świna River in order to connect the islands of Wolin and Usedom. Both of these investments are aimed at improving transport links for Świnoujście and the coastal belt with the rest of Poland within the international road network. The problem to be solved is the parking capacity of seaside towns during the holiday season, in particular, in Świnoujście and Międzyzdroje, with

the aim of improving their ability to accommodate the increased number of new one-day tourists eager for a seaside holiday. Due to the open borders between Poland and the other European Union member states, Germans are clearly concerned about the potential effects of overcrowding in the spa area as a result of the tunnel under the Świna River, which will facilitate the movement of vehicles from Poland. From the point of view of the spa function of this area, they expect an unfavorable increase in periodic car traffic and also a significant influx of visitors in resorts on the German side of the Bay of Pomerania [31,32].

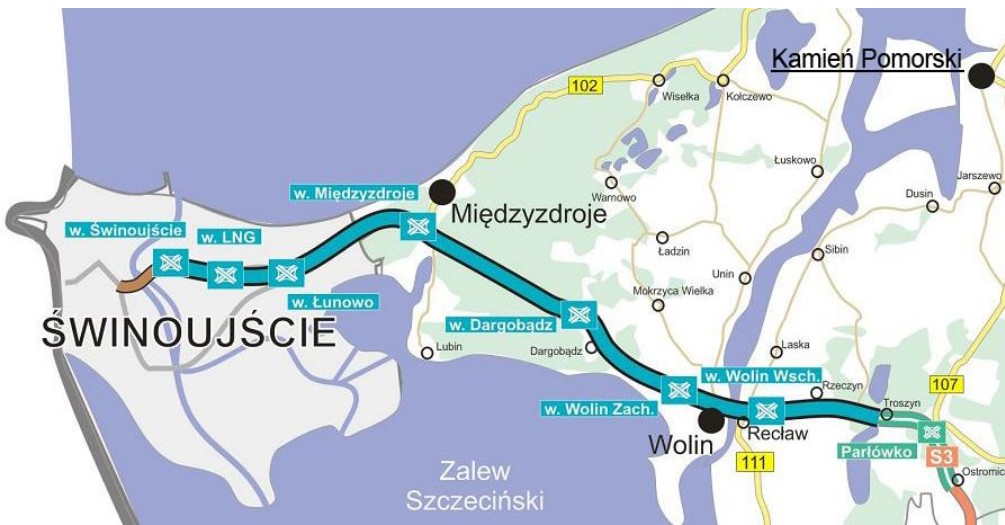

**Figure 6.** The course of the S3 expressway ending with a tunnel under the Świna River, which is to facilitate access and increase tourist traffic capacity in the direction of the city of Świnoujście.

The effects of large-scale investments are also visible in the area's balance of protected landscape areas. Over recent years, the size of protected landscape area has been reduced by 2296.55 ha (Table 1, Figure 7). This is due to the effects of economic development and the frequent issuing of permits for investments of supra-local and international importance under special acts, inconsistent with planning documents at the local and regional level, with environmental protection assessment studies and policy of sustainability.

### 3.1.3. Wind Farms

The development of wind farms has had a significant impact on the dynamics of landscape change in the area studied (Figure 8). There are 15 wind farms, with a capacity of 2 MW each, in the area of the Wolin commune. A wind farm covers an area of 225.2 ha. The further expansion of wind farms and their impact on the landscape in Poland was stopped by the Act of 24 April 2015, amending certain acts in connection with the enhancement of landscape protection tools (Journal of Laws of 2015, item 774). The basic terminological issue that decides the purpose, scope of research and the manner of conducting research is the definition of the landscape as an object influenced by wind farms. The essence of landscape has been the subject of numerous syntheses presented, among others, by Myga-Piątek [33] and Santos [34]. The most visible effect of the impact of wind power plants on the landscape is the reduction in the emission of harmful substances to the environment. The ecological effects of the construction of wind farms are: the noise that accompanies the operation of a wind turbine (mechanical and aerodynamic), the introduction of technical elements into the natural landscape (visual effect), the possible migration of animals from the areas where the power plant is located, threats to birds resulting from the possibility of a collision with the rotating blades of a power plant, the stroboscopic (flickering) effect, micro-vibrations and electromagnetic disturbances.

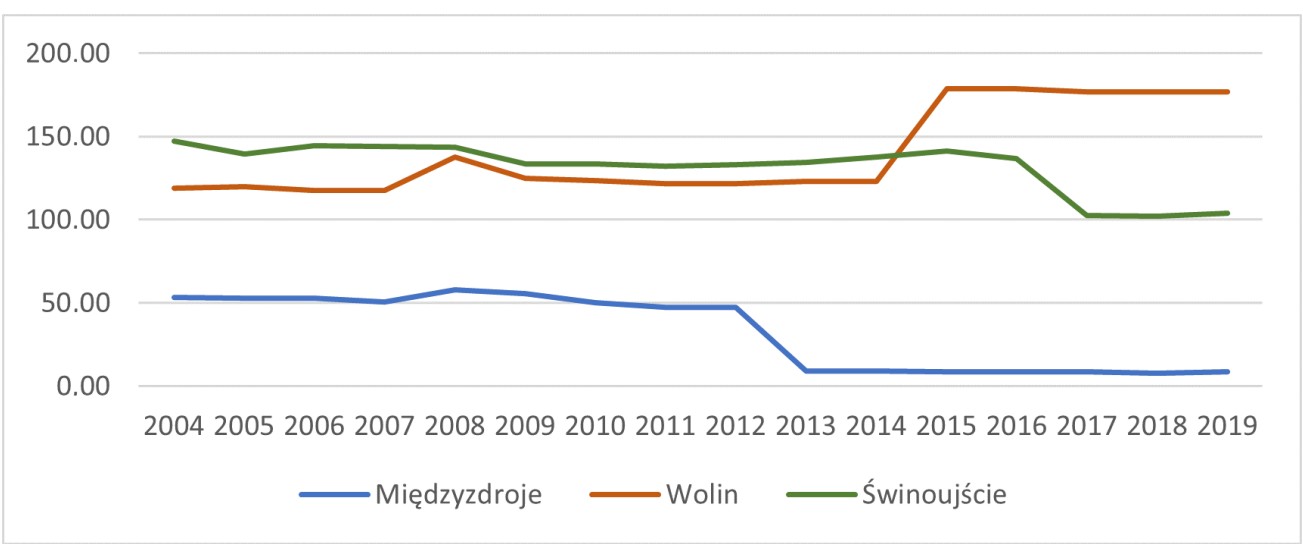

**Figure 7.** Analysis of the change in the area of forest land in hectares in the municipalities of Międzyzdroje, Wolin and Świnoujście in the years 2004–2019. Own study based on GUS data (accessed on 18 April 2021).

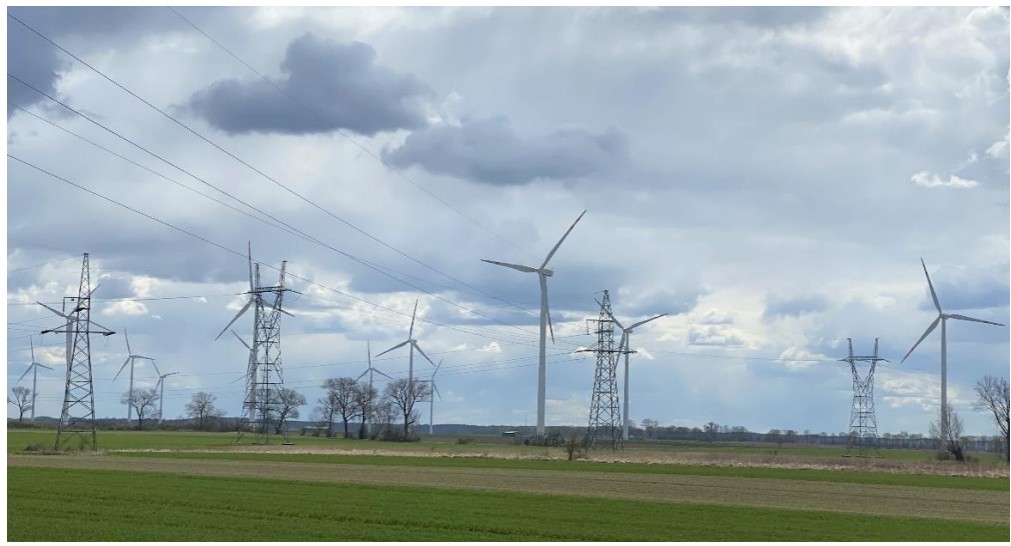

**Figure 8.** Wind farm near Wolin. The densification of energy infrastructure disrupts the traditional landscape of the rural area. Photo by Z. Paszkowski.

Wind power plants do not pollute the air, soil or water. However, it is often said that they cause "visual pollution" of the environment. This problem is all the more serious as the areas suitable for the construction of wind farms are usually coastal or mountain areas, the landscape values of which may be permanently affected by the construction of these installations. Wind power plants are tall devices (up to 150 m) with a color contrasting to the sky background and the earth's surface; therefore, their visibility from long distances is very clear. To assess the impact of the planned investments on the landscape, it is not enough to state that they are visible. Their influence on changing the current environment should also be considered, which is largely a matter of subjective perception, as it depends on the personal preferences of the evaluator.

The location of the research area creates the economic potential for wind energy. More and more investors are also noticing the great potential of offshore wind farms. The first offshore wind farms are to be commissioned on Polish sea waters from 2024 to 2033.

### 3.2. Natural Factors

The coastal zone is an extremely important natural and economic area of interaction between land and sea, while at the same time, the zone is extremely sensitive to all changes, both natural and anthropogenic. The coastal zone of the research area should be considered an area of clear conflict between economic development and the preservation of the natural landscape and existing geosystems. The dynamics of shoreline changes related to the landslides of cliffs and the destruction of dune-spit banks in the analyzed period were very high. In the analyzed period, as a result of natural factors, 75% of the cliffs and approximately 60% of the dune-sandbanks of the Baltic Sea were damaged [38,39].

Functional and spatial changes caused by human activity in an area of special landscape value, and affecting unfavorable landscape change, should be accompanied by compensation measures, restoring the possibility of landscape resilience. In this case, it would be important to shape the port pier adjacent to the beach and the dune landscape in such a way as to enable the renaturalization of this area (Figure 9).

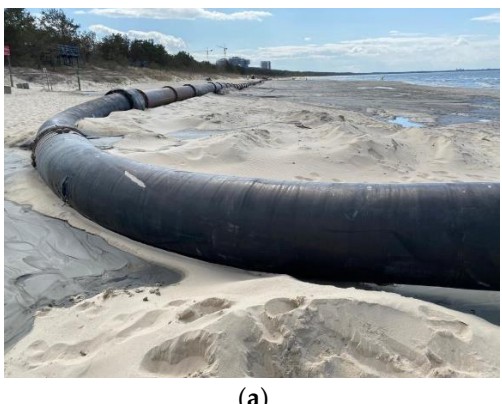
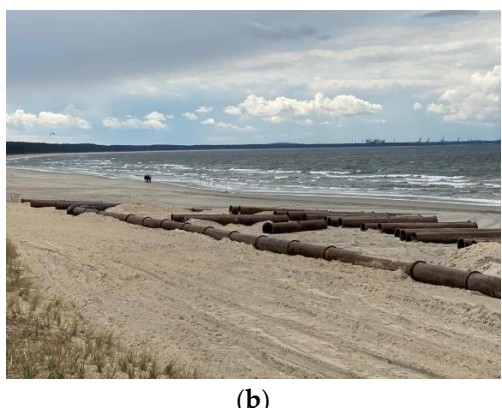

(**a**) (**b**)

**Figure 9.** Counteracting the decline in the surface of the beaches as a result of the rising water level in the Baltic Sea. The technical process of increasing the level of the beach in Międzydroje using sand from the excavation for the deep-water Świnoujście gas terminal. Structure to strengthen and raise the level of the beach (**a**) main pipe transporting sand to the beach; (**b**) a piping system to transport the sand to the beach before installation. Photo by Z. Paszkowski.

### 3.3. Blue-Green-Yellow-Grey Conflicting Areas

The West Pomeranian coast at the junction with the Bay of Pomerania is an area with many faces, functions and ranges of influence. There are various apparently incompatible functional areas, as well as phenomena and transformation processes, that must be taken into account in the system of integrated spatial planning.

The key problem is the polarization of development goals between the local society, the need to protect the landscape and natural environment resources, as well as the economic development goals of the state. Integrated planning in such a socially and environmentally sensitive landscape area causes conflicts of a social, legal, functional and spatial nature that are difficult to solve.

In the case discussed of the Baltic coast strip, based on the principles of sustainable development, there is a need to prioritize the preservation of the natural landscape and the extraordinary landscape values of the area of dunes and coastal beaches, pine and beech forest areas of the Wolin National Park and the Odra estuary areas, which are unique in terms of flora and fauna (Figure 10). These last factors were the reason the area was included in the Natura 2000 initiative. It is also protected as a drinking water resource. The part of the city of Świnoujście, located at the junction with the seaside wide beaches of the Bay of Pomerania, has been designated a Health Resort District, with special status in terms of maintaining air and water purity standards and restricting the spread of noise pollution.

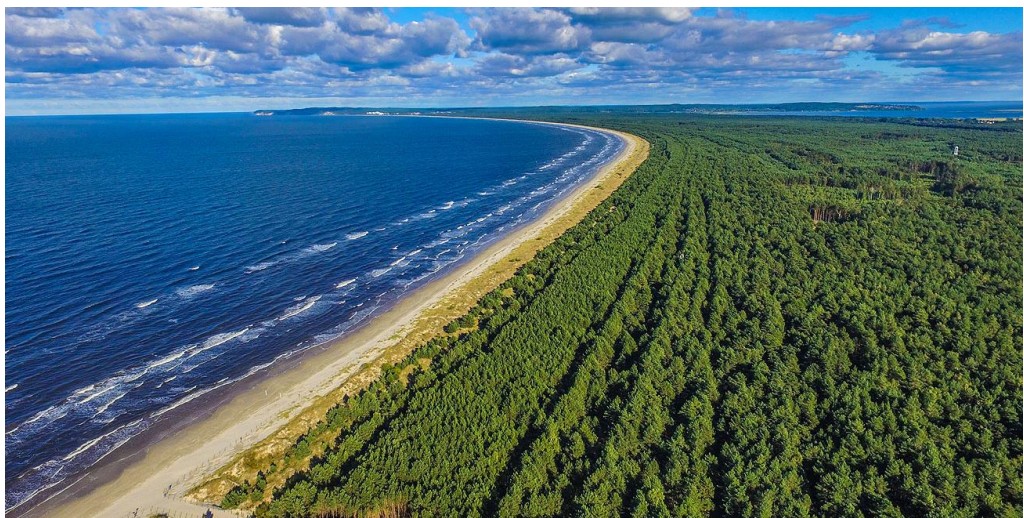

**Figure 10.** A view of the coast with sandy beaches between the port in Świnoujście and Międzyzdroje. This is an area that will be subject to significant investment changes in the near future, with dynamics depending on the economic situation. The development of port and guesthouse functions is planned in this area. The protection of the natural landscape of this part of the Bay of Pomerania is a particularly important factor for maintaining sustainable development. Photo by Daniel Szysz.

In contrast, the priority resulting from the economic development plans of this part of Europe indicates a different direction in development, with the expansion of the intermodal transport function and its necessary infrastructure.

In areas with a high polarization of development goals, the principles of integrated planning should be applied. The main goal of integrated planning is to jointly establish development priorities and ways of implementing them, with the assumption of minimizing adverse side effects. In order to better illustrate the emerging conflicts between particular priorities, a color code was adopted, assigning specific, characteristic colors to the various functions of this area (Figure 11).

Integrated planning in the landscape of the Odra estuary, at its confluence with the Bay of Pomerania on the Baltic Sea, covers four mutually competing areas:

- BLUE—seaside areas and water bodies,
- GREEN—areas protected in terms of nature,
- YELLOW—areas of tourist development, including buildings and complexes constituting cultural heritage
- GREY—areas for the development of multimodal transport (roads, railways, ports and ferry crossings)

There are certain relationships between these spatial units:

The BLUE and GREEN areas form a coherent ecosystem of priority importance to the area in question. Nevertheless, strong erosive processes take place within the coastal layer at the sandy coast. The YELLOW and GREY colors are areas of anthropogenic pressure, related to economic development. They are also mutually conflicting areas. The development of the spa function, although it benefits from convenience of transport, is not prepared to accept an increase in background noise resulting from the development of intermodal transport. The BLUE area is crucial for both YELLOW (tourism development) and GREY (intermodal transport development). Therefore, without a coherent, joint consideration of these areas and the mutual inter-dependence of areas of conflict, it is impossible to solve the problems generated by economic development in this area.

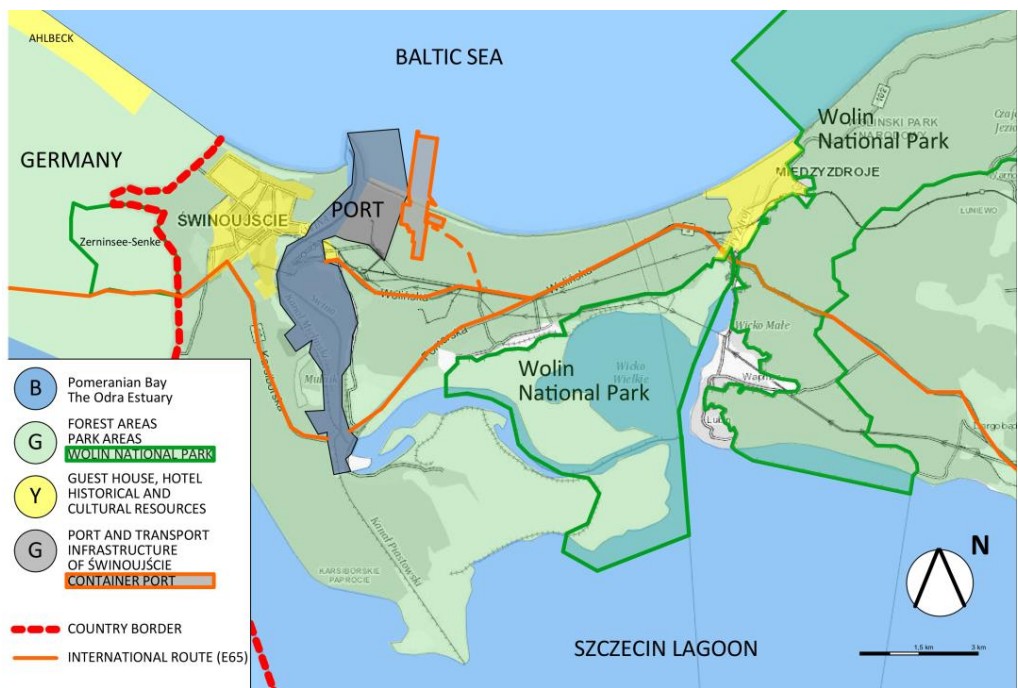

**Figure 11.** Diagram of functional distribution of conflict areas according to the BGGY color system. Comp. K. Krasowska.

Planning of a multidisciplinary, integrated nature should define development directions, as well as (or perhaps above all) take measures to prevent social conflicts, protect the natural environment and introduce various forms of environmental compensation—for the loss of value due to industrial development and tourism, taking into account the coastal specificity of climate change.

The coastal communes are aware of the need to care for the natural environment and are applying for compensation subsidies from investment promotors. Examples of applied environmental compensations in the studied area include the following activities:

- Construction of sewage treatment plants and modernization of existing treatment plants;
- Construction of multi-story car parks at the entrances to coastal cities and paid parking zones in order to relieve cities from the excess of motor vehicles during the summer and holiday seasons;
- Revitalization of the Spa Park in Świnoujście with amusement parks, an amphitheater, a rope park, playgrounds, walking paths in the dunes and the extension of the Promenade in Świnoujście—as compensation for deforestation and the attempt to focus tourist traffic in the "controlled" area;
- Construction of the Świnoujście–Międzyzdroje cycle path, as a section of the R10 Baltic cycle route, with wonderful panoramic views over the sea coastline and a difficult intersection with intermodal connections to the gas terminal and the planned container terminal;
- Reconstruction of the sandy beach in Międzyzdroje as an attempt to compensate for environmental damage and losses in the dunes (due to storms) and cliffs;
- and others.

However, the compensation works undertaken do not cover the damage done to the natural landscape due to the process of the disregard of environmental changes.

## 4. Discussion—Various Points of View of the Dynamics of Landscape Development

*4.1. The Dynamics of Landscape Changes and Sustainable Development*

"A landscape perspective fosters a multi-scale approach to sustainable landscape management and landscape planning. Additionally, a landscape scale is very useful for the innovative application of the common management paradigm to multiple uses in agriculture, forestry and water resource management" [35].

The applied research method of monitoring the dynamics of landscape changes is of particular importance to ecologically sensitive areas [36]. The Grey/Green/Yellow/Blue color system used in this method allows for a better visualization of the assessment of landscape changes taking place in the space and their dynamics. The assessment of the dynamics and directions of landscape changes depends on the adopted criteria and priorities. Assuming that the overriding value in the research area is economic development, and such an interpretation is mostly represented by supra-local authorities and some of the inhabitants of the research area, the high dynamics of anthropogenic pressure, illustrated in grey, can be considered a positive direction for changes.

The problem of the development of investments in the coastal belt results from the large, constantly growing demand for spending holidays at the Polish seaside and climate change [37]. On the one hand, this is due to the improvement of the economic situation in Poland, its opening up to neighbors from the West and South of Europe, their diminishing concerns about the quality of tourist services in Poland, the difficult international situation and the threats of terrorism in areas outside Poland that are considered to be tourist hubs, as well as issues related to the pandemic. On the other hand, the growing demand for residential premises in the coastal zone and the still existing possibilities of locating investments on the seafront, right next to the beach, have attracted large development companies, which see these investments as an opportunity to generate large profits. Examples illustrating these trends include the Radisson and Hilton hotel complex in Świnoujście, the Aquamarina skyscraper and the Wave apartment complex under construction in Międzyzdroje (Figure 12), the project to build two twin skyscrapers right next to the beach in Międzyzdroje (Figure 13a), and the gigantic Gołębiewski hotel in Pobierowo (Figure 13b).

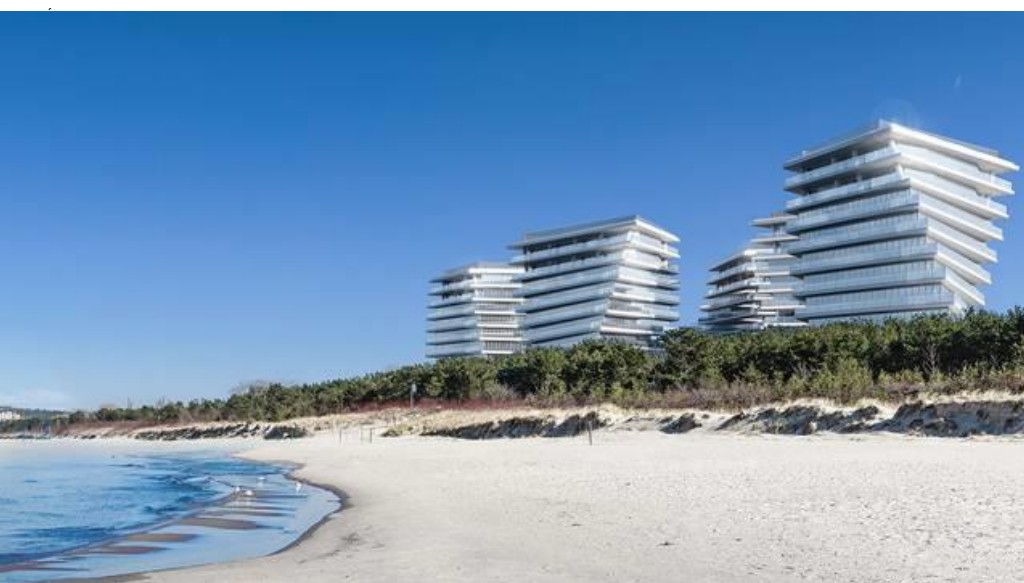

**Figure 12.** Completed project of the Wave Hotel and Apartment Complex in Międzyzdroje. These buildings, located above the dune strip, significantly change the skyline of the view of the shore of the Bay of Pomerania seen from the water side.

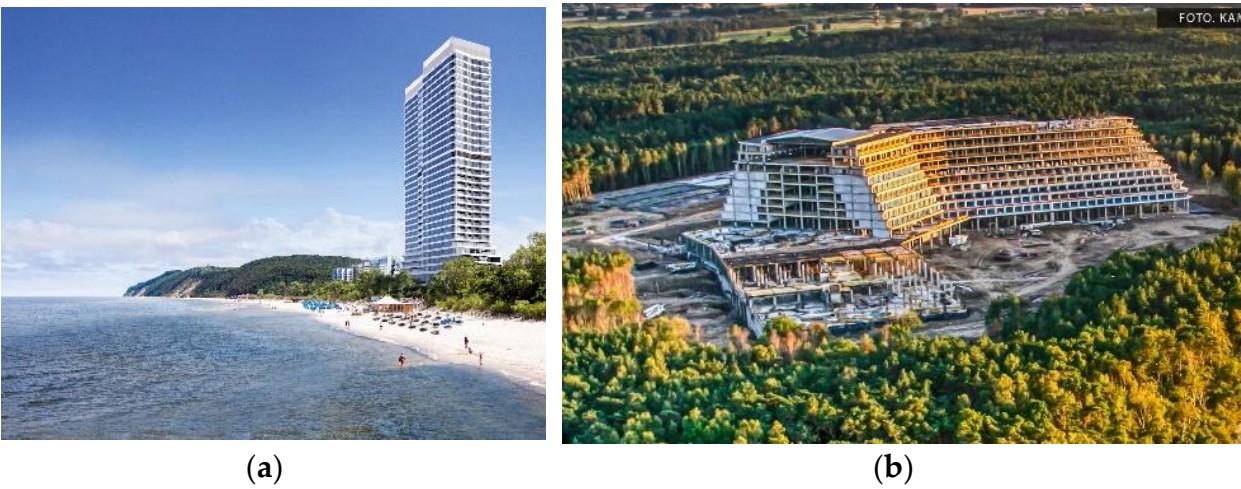

(**a**)      (**b**)

**Figure 13.** New development investments on the coastal belt. (**a**) The twin skyscrapers in Międzyzdroje—a project with building permission, (**b**) Gołębiewski Hotel in the village of Pobierowo—under construction.

There is a huge increase in the number of new flats and apartments for rent. At the same time, the number of permanent residents of the area under analysis is still decreasing. It follows that the new apartments are places of seasonal stay, which is mainly related to the development of tourism. A large number of these apartments are part of the short-term rental market—airbnb, booking.com, etc. [38,39] (Figure 14a,b and Figure 15).

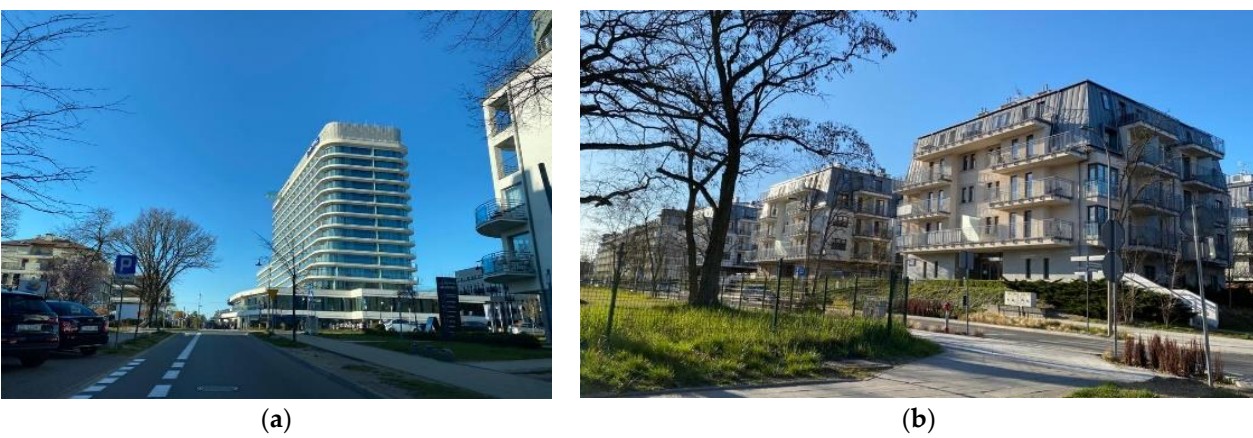

(**a**)      (**b**)

**Figure 14.** (**a**) Hotel Radisson, 15 floors in height, dominating the Świnoujście seaside, the highest building up to now in the research area, sets a new paradigm for seaside structures. (**b**) Guesthouse buildings in the area of the spa district of Świnoujście. Photos by Z. Paszkowski.

The same dynamics of landscape change can be assessed completely differently when the priorities of sustainable development and absolute protection of the natural environment are adopted. Here, for obvious reasons, preference is given to the low dynamics of landscape change, as well as maintaining proportionality between the development of industry, intermodal transport and environmental protection.

The thesis about the necessity of applying the principles of sustainable development, included in the Constitution of the Republic of Poland, does not exclude the possibility of economic development. However, the idea of sustainable development imposes an obligation to compensate for the loss of value to the natural environment, with pro-ecological actions in the form of environmental compensations, which should be included in those investments that are "worsening the conditions of the environment". In the case of the area

being analyzed, such projects include all intermodal transport investments currently being undertaken in this area [40,41].

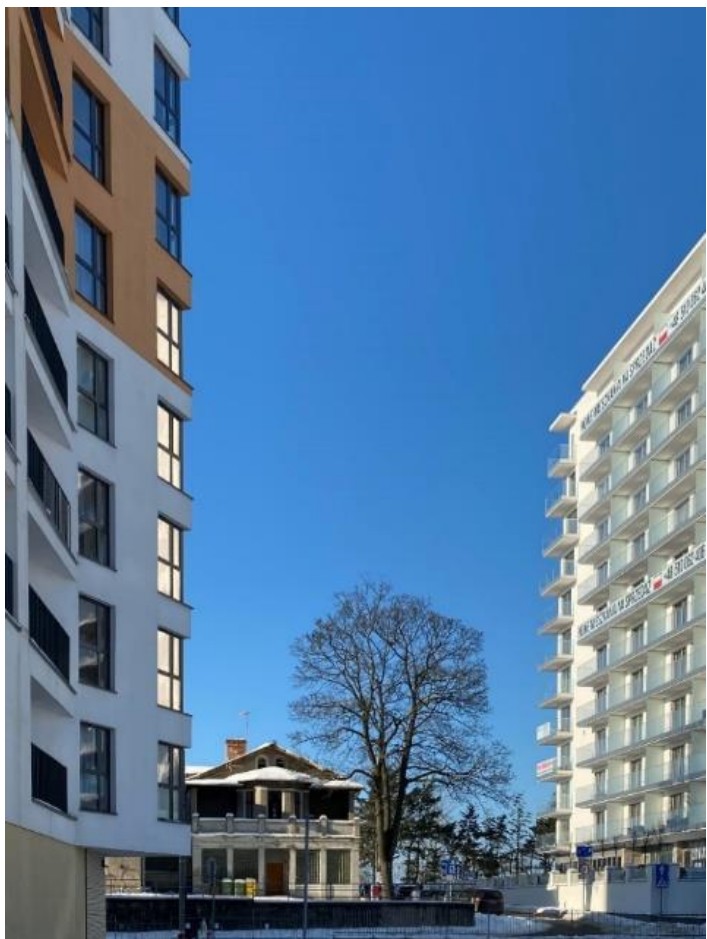

**Figure 15.** Międzyzdroje—the construction of new, multi-story individual recreation facilities has introduced a dramatic change in the scale of the existing, historically shaped, low guesthouse buildings. Photo by. Z. Paszkowski.

*4.2. Differentiation in Terms of Development Priorities*

The dynamics of landscape change are a measurable and visually clear symptom of whether the economic development process exceeds the acceptable limits of sustainable development. The dynamics of landscape change are the result of investment, cleaning and political activities. The anthropogenic pressure related to nature-protected areas, such as the coast of the Bay of Pomerania, is the result of the adopted policy of equalizing the disproportion in economic and "life standards" between Poland and Western European countries [25]. Recently, this economic factor-driven policy has been increasingly replacing the conservative environmental protection policy, which was dominant not so long ago. The effects of these changes in the attitude to the environment are visible in the unprecedented dynamics of landscape change [42].

*4.3. The Influence of Weather–Change on the Coast of the Bay of West Pomerania*

The average global sea level rise is accelerating significantly. Satellite data from the last 25 years clearly indicates that the acceleration of this growth is related to climate warming and the melting of the Greenland and Antarctic glaciers [22]. Progressive climate change, as well as natural factors on the coast, are changing the border between land and sea. The research area includes the Wolin National Park, with an almost twenty-kilometer-long coast of the Baltic Sea. A large part of the coastal landscape is an abrasive and scree cliff, which

is characterized by a sandy and gravel slope inclined at the angle of a natural chute. As a result, and additionally in the case of a parallel abrasion process, there is a continuous loss of the seashore and progressive erosion of the coast (Figure 16). This is influenced by the structure and morphology of the terrain, as well as by the number and strength of storms and precipitation. The consequence is a landslide, i.e., a sudden displacement of earth masses along the slip surface under the influence of gravity. In addition to natural factors, human activities can influence coastal erosion. Disturbances in the natural flow of sediments in the coastal zone as a result of, for example, the expansion of port breakwaters, shore fortifications and strips may aggravate the deficit of crumb material in the vicinity of the cliffs. The solution used to strengthen the pier with concrete blocks does not create such a chance and is an man-made example of the negative changes in the landscape (Figure 17).

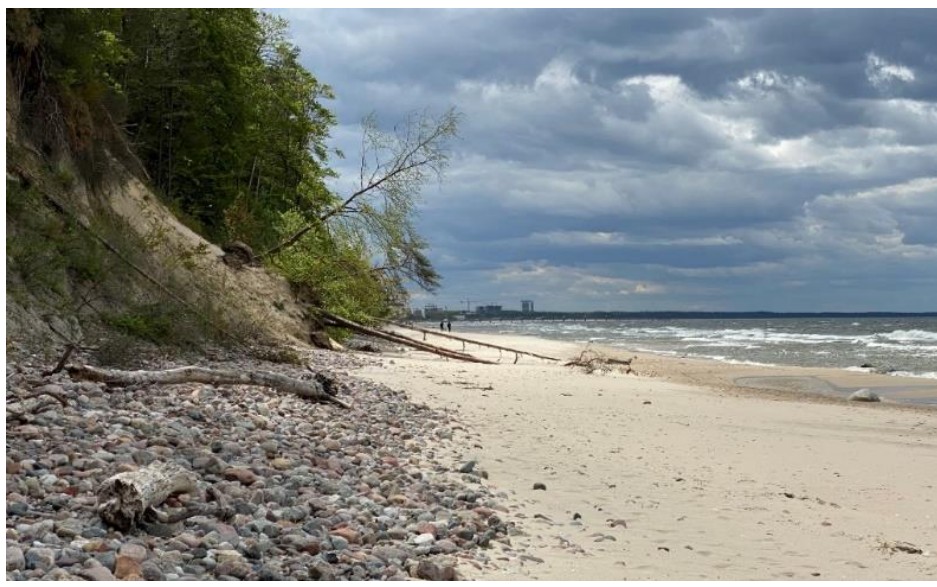

**Figure 16.** View of the characteristic displacement under the gravity influence of earth masses with trees along the slip surface at the Wolin National Park cliffs at the southern seaside of the Baltic Sea in Międzyzdroje commune area. Photo by Z. Paszkowski.

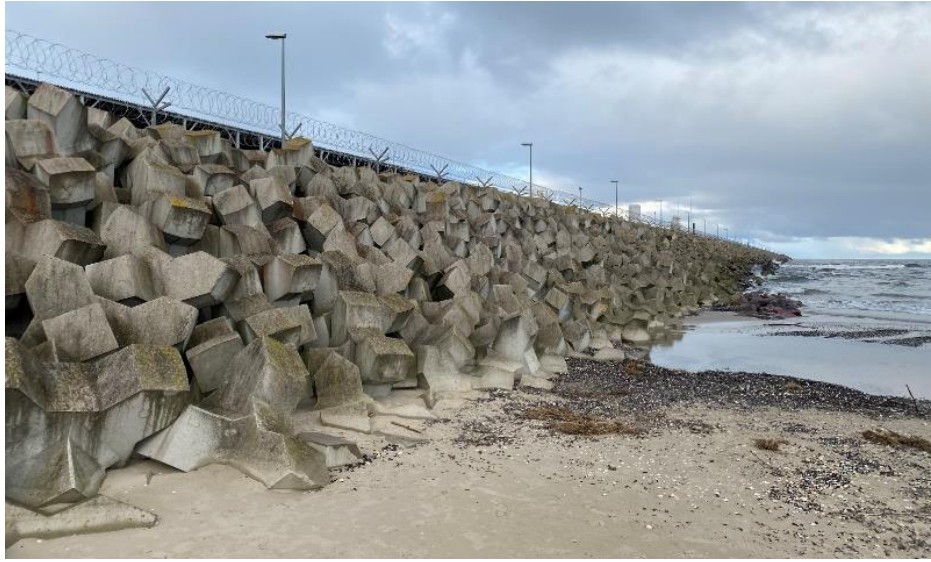

**Figure 17.** Świnoujście—the storm protection of the gas port pier changes the natural, soft dune landscape of the Baltic coast into a protective system of an unpleasant structure of concrete blocks. Photo by Z. Paszkowski.

## 5. Conclusions

Summarizing the research on the dynamics of landscape change in the area of the Baltic sea coast in the West Pomeranian Voivodeship, the following conclusions can be drawn:

- The dynamics of landscape change in the research area of the Bay of Pomerania on the coast of the Baltic Sea have undergone considerable intensity due to the strong anthropogenic pressure of tourism and the implementation of many investment tasks in a relatively short period of time. The following landscape changes are the result of direct and indirect human influences on the landscape.
- Human interference in the structure of the eco-system through deforestation, the construction of linear, transport and cuboid investments, especially the development of protected dune areas and of the buffer zone of the National Park, should be considered a direct impact.
- An indirect impact is the influence on the landscape in the studied area caused by human activity through water, soil and air pollution, changes in the level of groundwater and its composition, and changes in plant cover and land topography.
- The natural landscape of the southern Baltic coast in the area of the Bay of Pomerania has been increasingly transformed into a cultural landscape, in which cubature and infrastructure objects have become landscape dominants.
- The dynamics of landscape transformations are a visual measure of the socio-economic changes taking place. They also serve as a warning against excessive human interference in natural, protected areas, due to their non-reproducibility and importance for the ecosystem.
- Economic development is hard to stop. However, it is about maintaining proportion and moderation, ensuring environmental compensation through new plantings of greenery and reconstruction of beaches damaged by erosion resulting from the construction of non-natural barriers to the free flow of coastal waves and climate change. It is important to limit construction, which brutally interferes with the existing landscape and changes its essential qualitative features at the cost of having a nice view from a window.

The research problems outlined in this article require further in-depth, interdisciplinary study, especially because the effects of the changes currently being introduced will be noticeable in the longer term. The authors are convinced that the method of monitoring landscape change through the analysis of the causative factors mentioned in the article will allow for an objective assessment of the observed changes. Witnessing the monitoring of dynamic changes in the landscape, the authors would encourage authorities at different levels to undertake sustainable, synergistic landscape management as one of the most important factors in the complex, multi-sectoral, multi-spatial planning of integrative, sustainable landscape development.

**Author Contributions:** Conceptualization, Z.W.P. and K.K.; methodology, K.K. software, K.K.; validation, Z.W.P.; formal analysis, Z.W.P.; investigation, K.K., Z.W.P.; resources, K.K.; data curation, K.K.; writing—original draft preparation, Z.W.P. writing—review and editing, K.K., Z.W.P.; visualization, K.K., Z.W.P.; supervision, Z.W.P.; project administration, K.K..; funding acquisition, Z.W.P., K.K. All authors have read and agreed to the published version of the manuscript.

**Funding:** This research received no external funding.

**Institutional Review Board Statement:** Not applicable.

**Informed Consent Statement:** Not applicable.

**Data Availability Statement:** Data supporting reported results can be found at: https://bdl.stat.gov.pl/BDL/start.

**Acknowledgments:** The developed research topic is the study subject at the Department of History and Theory of Architecture at the Faculty of Architecture of the West Pomeranian University of

Technology in Szczecin as part of our own research. The authors would like to thank everyone who helped with the data and visual materials needed for the research and development of this article during the difficult times of the pandemic.

**Conflicts of Interest:** The authors declare no conflict of interest.

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
