# Peer review of "The Need to Maintain Sustainability in the Dynamic Anthropogenic Changes in the Natural Landscape of the Bay of Pomerania in Poland"

_sustainability, doi:10.3390/su15031928_

Round 1

Reviewer 1 Report

This article presents research on the dynamics of landscape changes of the Pomeranian Bay on the south-western coast of the Baltic Sea, taking into consideration the impact of the development of the port in ÅšwinoujÅ›cie harbour. Analyze the influence on landscape features from both natural conditions and human factors, with clear research objectives and in-depth analysis. The objectives of this study are  appropriate for this journal. However, there are still some major Issues to further illustrate the manuscript.

1. The articles mainly focus on qualitative analysis of landscape dynamics, lacking quantitative analysis, especially the lack of quantitative analysis of landscape dynamics and influencing factors

2. Could the author please indicate how visual pollution affects the landscape?

3. Where is the figure4? I cant see in the MS

Author Response

Reviewer 1

English language has been improved, but if there is still the further improvement needed, we will kindly ask to do it by the MDPI language service.

Is the content succinctly described and contextualized with respect to previous and present theoretical background and empirical research (if applicable) on the topic? The content description has been contextualized with respect to theoretical background, empirical research and observations have been conducting in the recent years. 

Are all the cited references relevant to the research?
Yes, the relevance has been improved.

Are the research design, questions, hypotheses and methods clearly stated?
The research design, questions, hypotheses statements have been improved.

Are the arguments and discussion of findings coherent, balanced and compelling?
The arguments, discussion of findings has been improved  according to the Reviewers  suggestion, and seems to be now more balanced and compelling.

For empirical research, are the results clearly presented?

For empirical research there are statistic data used, which enrich and justify the observations of local processes. These observations are not only superficial, but are based on the involvement of the authors in planning and design processes in the described area.

Is the article adequately referenced?

The number of article references has been enlarged significantly, what improved the scientific level of the article.

Are the conclusions thoroughly supported by the results presented in the article or referenced in secondary literature?

The conclusions are in higher grade /mainly/ supported by the results of author’s research and additionally referenced in the secondary literature listed in the article.

  1. The articles mainly focus on qualitative analysis of landscape dynamics, lacking quantitative analysis, especially the lack of quantitative analysis of landscape dynamics and influencing factors. The liable quantitative analysis needs a lot of data from a long period of time. Some quantitative figures has been given in the data tables. There were however no appropriate instruments at the present time available to the authors to measure all the listed natural and man-made landscape changes in a form of comprehensive and complex system. The article stresses the need of landscape change dynamics monitoring in order to maintain landscape natural values. The main landscape changing influencing factors are listed in the article.
  2. Could the author please indicate how visual pollution affects the landscape? Visual pollution affects the landscape mainly by inappropriate impact of man-made interventions into the landscape, what destroys the environmental balance of sustainable capacity and harms the natural beauty with oversized buildings, infrastructure facilities and artificial visual components accompanying the human settlements and infrastructure.
  3. Where is the figure4? I can’t see in the MS. The figure 4 has been added to the revised version.

The authors would like to thank the Reviewer 1 for the valuable comments, and hope, that the text is now more clear and that the improved version of the article will be suitable for publishing.

With best regards

Katarzyna Krasowska, Zbigniew W. Paszkowski

Szczecin, 28.12.2022

Reviewer 2 Report

Please see attached pdf for comments.

Author Response

Reviewer 2

English language has been improved, but if there is still the further improvement needed, we kindly ask to do it by the MDPI language service.

Is the content succinctly described and contextualized with respect to previous and present theoretical background and empirical research (if applicable) on the topic? The content description has been contextualized with respect to theoretical background and empirical research and observations conducted in the recent years. 

Are all the cited references relevant to the research?

Yes, they are.

Are the research design, questions, hypotheses and methods clearly stated? The research design, questions, hypotheses statements have been improved.

Are the arguments and discussion of findings coherent, balanced and compelling? The arguments, discussion of findings has been improved and seems to be now more balanced and compelling.

For empirical research, are the results clearly presented?

For empirical research there are statistic data used, which enrich and justify the observations of local processes. These observations are not only superficial, but are based on the involvement of the authors in planning and design processes in the described area

Is the article adequately referenced?

The number of article references has been enlarged significantly (up to 45 positions).

Are the conclusions thoroughly supported by the results presented in the article or referenced in secondary literature?

The conclusions are in higher grade supported by the results of author’s research and additionally referenced in the secondary literature listed  in the article.

Responses to the General comments of the Reviewer 2

The abstract should give more context for the study; consider adding two sentences about the requirement for this research and the state of development in Pomeranian Bay. Abstract have been improved by changing the entire text.

Research aims and objectives should be stated explicitly in the Introduction. Consider being more specific about these in the final paragraph of the Introduction.  
Research aims and objectives have been included in the Introduction.

The methods are vague and no specific methodologies are referred to.
The methods has been described in the Chapter 2 of the article.

The author should provide the reader with what databases were used to extract data, for example, in line 114, the author refers to “the analysis of statistical data” but does not state where that data was from.
The reference to the cited data has been included in the article.

The Results section included no statistical analysis regarding landscape change. This should form an integral part of the study.
The Result Section has been provided with statistical analysis.

This Discussion is very small, and this is due to the fact much of the discussion has taken place in the Results. Results should be stated in a more concise and clear way, and much of the text should be moved to the Discussion.  
The /structure has been improved/ texts have been moved accordingly.

A lot of the discussion that was included in the Results was great, and should be elaborated on in the Discussion.
The texts has been moved accordingly to the suggestions of the Reviewer 2.

Readers should be able to understand what is happening in the Figure/Table without having to read the text, so Table and Figure legends should be made more detailed.
The Figure’s descriptions have been extended in order to be able to understand the merit of the Figures.

There are only 25 references. More should be added to bolster the author’s results.
20 more references have been added. There are now 45 references.

Line by line

Line 86: Must detail what software was used to produce this map and reference the software. Described. The information has been added.

Line 129: This figure does not help the reader to visualize the methods. Consider changing how this is presented. Potentially, this does not need to be visually represented, maybe a table would be better?

The Figure has been re-modelled.

Line 145: This should be in the Methods, not the Results. It has been moved.

Line 152: The figure this caption is referring to is not in the manuscript. The missing Figure has been added.

Line 157-175: Much of this paragraph should be moved to the Discussion as the author is discussing the implications of their results.
The text of this paragraph has been moved to Discussion chapter.

Line 201: Can the author provide more detail about the trees that were planted following this development? Reforestation is often thought to be successful without considering whether native tree species were planted or whether it is the right habitat for particular tree species.

The proper text has been added. The reforestation planned and performed in the area of investigation is based only on native tree species, like: pines, oaks, birches and beeches.

Line 214-225: This paragraph is necessary context to the methods of the study; this would be better suited in the Introduction/Methods [study site]. It has been moved.

Line 284-333: This section could be a lot shorter, and much of this text should be moved to the Discussion, e.g., 320-326 is proposing a potential solution. It has been moved.

The authors would like to thank the Reviewer 2 for valuable comments, and hope, that the text is now more clear and that the improved version of the article will be suitable for publishing.

With best regards 

Katarzyna Krasowska, Zbigniew W. Paszkowski

Szczecin, 28.12.2022

Reviewer 3 Report

The article constitutes a relevant contribution to the theme of the journal, focusing on a specific case study such as the Pomeranian Bay on the south-western coast of the Baltic Sea. The text is well structured and the sections are used correctly.

The arguments are clear and coherent. The conclusions are expressly stated and justified. The bibliography is appropriate to the objectives and theme of the article. The images illustrate the content of the article.

Author Response

Reviewer 3

There is no comments or remarks in the Review 3. 

We would like to thank cordially the Reviewer 3 for the high review score  of our article, which as we hope, will be suitable for publication soon.

Best regards

Katarzyna Krasowska and Zbigniew W. Paszkowski

Szczecin, 28.12.2022

Round 2

Reviewer 2 Report

Please find the comments in the attached pdf.
